# The Stabilizing Effect of Carboxymethyl Cellulose on Foamed Concrete

**DOI:** 10.3390/ijms232415473

**Published:** 2022-12-07

**Authors:** Yongcheng Ji, Qijun Sun

**Affiliations:** School of Civil Engineering, Northeast Forestry University, Harbin 150040, China

**Keywords:** foam concrete, CMC concentration, foam stability, compressive strength, pore structure

## Abstract

Foam concrete is widely used for its excellent properties, such as light weight, heat insulation, fire resistance, and sound insulation. The stability of foam is the main factor that affects the mechanical performance of foam concrete. The experiments are designed from two perspectives: the foam’s stability performance and the foam concrete’s modification effect. The effects on foam volume, foam half-life, foam bleeding rate, and foam pore size were investigated based on different concentrations of foam stabilizer CMC (0%, 0.1%, 0.2%, 0.3%, 0.4%, 0.5%). A combination of macroscopic testing and microscopic analysis, a comparative study of dry density, water absorption test, mechanical property test, and pore structure analysis were conducted after using the modified foam for foam concrete. It is shown that the addition of CMC has an enhanced effect on foam stability. Foaming volume, water secretion rate, and average pore size showed a decreasing trend with the increase of CMC admixture, while the foam half-life displayed an increasing trend. In addition, adding CMC reduces the dry density and improves water absorption and compressive strength. The pore structure development of foam concrete has a noticeable improvement effect, and the optimal amount of admixture is 0.4%. Research results provide a reference for applying thickening foam stabilizer CMC in foam concrete.

## 1. Introduction

Foam concrete (FC) is material prepared by physical means, such as rapid mixing or compressed air. The slurry of foam and cement-based materials are then mixed and stirred. The lightweight foam concrete of different densities can be prepared by controlling the foam preparation [1,2,3]. In addition, foam concrete contains closed pores, which gives it the advantages of being lightweight and giving good heat insulation, sound insulation, fire prevention, and seismic resistance. Foam concrete is applied to wall masonry materials, particle boards and blocks, which play roles in wall insulation, pipe insulation, and frame-filled walls [4,5,6]. However, its low density often leads to disadvantages such as low strength, poor stability, high water absorption and shrinkage, and easy cracking [7]. Therefore, strengthening the stability of lightweight foam concrete is a hot topic of current research. 

Foam is easy to combine and break in the preparation process, so it is crucial to improve its foam stability [8]. Therefore, adding a foam stabilizer is usually used to improve the stability of foam concrete materials. Typical foam stabilizers can be divided into three categories according to their mode of action. Firstly, synergistic foam stabilizers enhance the interaction between surface adsorption molecules through synergistic action [9], making the foam film more elastic and less permeable, thus improving foam stability. Secondly, viscosity-increasing foam stabilizers increase the viscosity of the liquid phase to reduce the discharge rate of foam and prolong the half-life [10], thus improving foam stability. The third category is the solid particle–type stabilizer [11]. It improves foam stability by irreversibly and spontaneously adsorbing solid particles at the gas-liquid interface, balancing the interfacial tension, forming a solid-liquid-gas three-phase foam, impeding the transfer of the gas-liquid phase, and reducing the physical drainage between the gas-liquid phase. 

Cellulose ether is common to water retention and thickening agent for cementitious materials. It includes hydroxyethyl cellulose, hydroxypropyl cellulose, hydroxymethyl cellulose, and carboxymethyl cellulose. CMC belongs to anionic cellulose ethers, cellulose-like substances with carboxymethyl substituents that are easily dispersed in water to form transparent colloidal solutions and have a high thickening efficiency due to the abundance of hydroxyl groups as well as hydrophobic chain segments [12]. Acting in foam concrete, they can enhance the denseness of the foam surface layer, improve the ability of bubbles to resist rupture, and eliminate pore connectivity, rupture, and escape so that bubbles can exist stably in the cement slurry for a long time [13].

Researchers have studied the application of CMC in cement-based materials. Farooque, K. N. [14] added CMC of different concentrations to ordinary Portland cement (OPC). By comparison, it was found that the water absorption and porosity of OPC-CMC cement decreased, and fiber-type substances were formed in hardened cement paste. Mishra, P. C. [15] used FT-IR and XRD technology to analyze the phase of materials and studied the interaction between surface CMC and cement hydration products. The addition of CMC reduced the water absorption and enhanced the corrosion resistance. Lu, Ziyi [16] found that there are many membrane substances in the hardened cement paste bound with CMC, which are well combined with the hydration products. Shi, Xiuzhi [17] studied the fluidity and strength characteristics of foam cement paste backfill material (FCPB) and found that the optimal combined concentrations of sodium dodecyl sulfate and CMC were 0.5 g/L and 0.2 g/L, respectively. Merachtsaki and Domna [18], who studied the essential characteristics of foam and surfactant through the experimental system design based on the response surface method, found that the choice of surfactant and CMC concentration has a significant impact on the foam characteristics of foam concrete and then affects the pore-size distribution of foam concrete. Research by Sahu and Sritam Swapnadarshi [13] shows that adding 0.2% CMC to 2.5% sodium sulfate solution can effectively improve foam quality, and foam discharge and large-size foam can be significantly reduced. At the same time, the compressive strength of foam concrete increased by 20%. In addition, adding CMC to a surfactant solution improves the surfactant’s viscosity and the foam’s microstructure [19]. 

The above research mainly focuses on the macro performance of cement-based materials. However, the research on foam concrete needs to be more comprehensive, and the microanalysis of foam liquid after adding foam stabilizer CMC is less. Therefore, foam concrete with a dry density of 800 kg/m^3^ was prepared in this study. Sodium dodecyl sulfate (SDS) is used as the foaming agent and sodium carboxymethyl cellulose (CMC) as a foam stabilizer. (i). The concentrations of foam stabilizing agent CMC (0%, 0.1%, 0.2%, 0.3%, 0.4%, 0.5%) were compared in groups to explore the influence of CMC on foam performance under different concentrations. (ii). The foam with CMC was applied to the foam concrete. (iii). Through the macroscopic test of the dry density, water absorption, and compressive strength of the foamed concrete, the pore structure analysis, and the microscopic analysis, the foam stabilization effect of CMC and its influence on the performance of foamed concrete was studied.

## 2. Results and Discussion

### 2.1. Foam Stability Performance Test

#### 2.1.1. Foam Volume and Foam Half-Life Analysis

The foaming volume and the half-life of foam measure the stability of foam and the foaming capacity of the foam solution. The longer the half-life, the better the stability of the foam [20]. The influence of CMC concentration on foaming volume and half-life of foam is shown in Figure 1.

Figure 1 shows that the foaming volume of the solution gradually decreases with the increase of CMC concentration, while the foam half-life shows an apparent rising trend. Compared with the control group, the foam volume decreased by 12.2%, 24.3%, 33.7%, 40.8%, and 46.9%. When the CMC concentration was 0.1–0.5%, the foam volume decreased by 31.6% on average. The analysis shows that adding CMC has a specific inhibitory effect on the foaming volume of the solution. At the same time, the growth rate of the half-life bubble can be divided into two stages according to the data in the figure. The growth rate of the foam half-life was faster when the mass fraction of CMC was in the range of 0–0.40%; from 7.5 min to 48.5 min, the increase was 41 min, and the magnitude of increase was 5.4 times. However, it was found that the foam half-life grew more slowly from 48.5 min to 51.5 min, with an increase of 3 min and 0.06 times when the mass fraction of CMC was in the range of 0.40–0.50%. The analysis shows that incorporating CMC can significantly improve foam half-life, thus improving foam stability, but there is a turning point that slows down the growth rate. In this section, the concentration of CMC is 0.4%.

#### 2.1.2. Analysis of Foam Bleeding Rate

Due to the internal and external pressure difference and gravity effect of foam liquid film, foam always tends to drain, which will reduce the thickness of the foam liquid film. With liquid discharge, foam continues to merge, become more prominent, and finally breaks when the liquid film strength is insufficient [21]. Therefore, the delay of the foam bleeding rate has a crucial role in the stability of the foam. The foam bleeding rate in each group as a function of time is shown in Figure 2a. The foam bleeding rate in 30 min and 60 min in each group is shown in Figure 2b.

Figure 2a shows that the increase of the foam bleeding rate is slow when time is less than 20 min. Because the foam film tension can keep the balance of internal and external pressure, the film thickness does not change significantly, and the amount of precipitated solution is small. However, foaming water secretion increases faster when time is longer than or equal to 20 min, and the 0% CMC addition has achieved 26 mL at 60 min, which is as high as 23 mL compared with 10 min. It can be explained that the internal pressure difference in broken foam reduces the thickness of the liquid film. The merging and rupture between the foams, leading to increased water secretion, is the cause of this phenomenon [22]. Thus, the volume of water secretion will increase over time, and the foam stability will become less and less stable. On the other hand, the foam bleeding rate decreased with the increase in CMC concentration. At 60 min, the 0.4% CMC concentration reached 15 mL, a difference of 11 mL compared with the 0% CMC concentration. Figure 2b shows that the foam bleeding rate decreased when the CMC concentration was less than 0.4%. The foam bleeding rate decreased by 54.1% and 57.6% compared to the control group at 30 and 60 min, respectively, with the CMC concentration equal to 0.4%, and the foam bleeding rate showed a flat trend when the CMC concentration was more significant than 0.4%.

The analysis shows that the foam bleeding rate increases with time; the foam bleeding rate is most intense at 60 min, and the stability is the lowest. However, adding CMC dramatical reduces the foam bleeding rate simultaneously, which can effectively slow down the rate of a foam burst. The effect is best when less than 0.4% is present in the admixture.

#### 2.1.3. Microscopic Analysis of Foam

The influence of CMC concentration on foam size is studied and the microscopic changes of each group at 60 min is studied. Microscopic pictures of each group at 60 min are shown in Figure 3, and the average size of bubbles in each group is shown in Figure 4.

Figure 3 shows that foam consists of many tiny bubbles; the number of bubbles under the same vision is more with the constant increase of the concentration of the foam stabilizer CMC. The surface bubbles are more closely distributed than the control group, and the number of tiny bubbles is also increasing. The analysis shows that adding CMC reduces the internal bubble burst of foam concrete at the same volume, leading to an increase in the number of internal bubbles. Figure 4 shows that the average size of each group of bubbles is 200–1000 μm. The average size of bubbles shows a significant downward trend when the CMC concentration is less than 0.4%, and the average pore size of bubbles tends to be flat when the CMC concentration is higher than or equal to 0.4%. The average size of bubbles in the control group was the largest, at 987 μm. The average size of bubbles in the C0.5 group was the most minor, at 223 μm, and the decrease was 77.4%.

The reduction of the average-size bubbles is due to the addition of CMC to the foaming solution, which improves the solution viscosity [13], so improving the strength of the liquid film [23], further inhibiting the combination and rupture of bubbles, effectively alleviating the generation speed of large bubbles, and giving foam a specific stability.

### 2.2. Foam Concrete Performance Test

#### 2.2.1. Dry Density and Water Absorption Test

Figure 5a shows the influence of CMC concentration on foam concrete’s dry density and water absorption. Figure 5b shows the fitting graph between dry density and water absorption of foam concrete.

Figure 5a shows that the dry density of the foam concrete gradually decreases, and the water absorption rate progressively increases with the addition of the foam stabilizer CMC. It has a dry density of 893 kg/m^3^ and a minimum water absorption rate of 10% at a concentration of CMC 0%. The dry density of the foam concrete was 788 kg/m^3^ with a stabilizer concentration of 0.5%, which is a reduction of 115 kg/m^3^ with a higher water absorption rate of 19.6%, which is an increase of 9.6% compared to C0.

In order to further analyze the correlation between dry density and water absorption in foam concrete, the measured dry density and water absorption test data of foam concrete are fitted. Figure 5b shows that polynomial fitting is adopted, with a high degree of fitting and good correlation. Here x is for dry density, and y stands for water absorption. The water absorption of foam concrete decreases with the increase in dry density. The higher the dry density is, the lower the water absorption is.

#### 2.2.2. Compressive Strength Test

Figure 6a shows the relationship between the concentration of CMC and the compressive strength. Figure 6b shows the fitting graph of CMC concentration and Compressive Strength.

Figure 6a shows that the compressive strength of foam concrete increases first and then decreases with the increase in the concentration of foam stabilizer CMC. For example, the compressive strength of C0.1 was 5.4 MPa, while the compressive strength of C0.4 was the highest (6.5 MPa), which was increased by 20% and 44.7% compared with the control group C0. Although the compressive strength of C0.5 is slightly reduced to 6.26 MPa, it is also significantly increased compared with the control group. It is because the increased foam stability by CMC and the thickness of the foam liquid film lead to the rupture of the foam in the cement slurry being mitigated. In addition, the pore size of the bubbles becomes more refined and uniform, which improves the compressive strength of the foam concrete [19]. However, with the increase of CMC admixture, the foam content inside the slurry increases and the porosity becomes larger, reducing the strength of the foam concrete, and explaining the reduction in compressive strength present after reaching the peak.

In order to better confirm the correlation between the CMC concentration of foam concrete and the compressive strength, the test data of the compressive strength of foam concrete were fitted. As shown in Figure 6b, a polynomial fitting was used, with a high degree of fitting and good correlation, where x is for CMC concentration and y represents compressive strength.

Figure 7 shows the compression process of foam concrete sample as a function of time. At the initial stage of loading, the relative load is small, and there is no obvious crack on the sample surface (as shown in Figure 7a,e). With the gradual increase of the load, the stress is concentrated on the weak part inside the sample, and on the sample surface there start to appear small cracks (as shown in Figure 7b,f). As shown in Figure 7c, several long and thin cracks appeared on the surface of the sample at 52s, some of which were through cracks. As shown in Figure 7g, an obvious through crack appeared on the surface of the sample at 58s, and the sample broke away from the elastic stage and entered the plastic damage stage. However, after 1 min, the samples lost their bearing capacity. In Figure 7d, the failure mode was layer-by-layer peeling, and a large number of cracks appeared. In Figure 7h, the cracks were further extended, and there was a crushing area in some parts. Through the surface observation and time comparison of the two groups of samples, it was found that the number of cracks in the group without CMC is more than that in the group with CMC, and the time to lose the bearing capacity is shorter. The analysis shows that CMC inhibits the generation of cracks in foam concrete and improves the bearing capacity, which corresponds to the results of a compressive strength test.

#### 2.2.3. Pore Structure Analysis

Generally, the indexes to evaluate the pore structure of foam concrete in experiments are pore-size distribution, average pore size, porosity, and roundness [24].

Figure 8 shows the pore-size distribution of foam concrete at different CMC concentrations. As shown in Figure 8a, the pore size of foamed concrete is mainly distributed in the range of 200–800 μm, and the pore size range varies with different CMC concentrations. The percentage of pores of a size less than 200 μm increases with the increase of CMC concentration, and the percentage is small. The percentage of pore sizes of 200–400 μm increases with the increase of CMC concentration, and C0.5 was 52.7% compared with C0. The percentage of pores of sizes of 400–600 μm decreases with increasing CMC concentration, and C0.5 is 20.3% compared with C0. The pattern of a hole size greater than 600 μm is the same as that of a hole between 400 and 600 μm. The percentage of 600–800 μm size pores decreases from 46.8% of C0 to 19.7% of C0.5. The percentage of pore sizes of 800–1000 μm decreases with increasing CMC concentration, and C0.5 was 0.8% compared with C0. When the CMC concentration is more significant than 0.3%, the percentage of the pores decreases to 0.0%, and the pores above 1000 μm disappear. Figure 8b shows that the concentration of CMC is 0%. The percentage of pores of a size of less than 200 μm is 0.7%, from 200–400 μm is 3.2%, 400–600 μm is 42.5%, 600–800 μm is 46.8%, 800–100 μm is 5.3%, and greater than 1000 μm is 1.5%. With the increase of CMC concentration, the percentage of pores of sizes less than 400 μm (0–200, 200–400) increases rapidly, while the percentage of pores of sizes greater than 600 μm decreases significantly. The CMC concentration is 0.4%, the hole size is less than 1000 μm, the CMC concentration is 0.5%, and the percentage of pores smaller than 400 μm in size is 59.2%. It can be seen that the increase in CMC concentration makes the pore size smaller and smaller, and the pore size changes gently when the CMC concentration is greater than or equal to 0.4%.

Figure 9 shows the foam concrete’s average pore size at different CMC concentrations. It can be observed that the average pore size of foam concrete decreases with the increase in CMC concentration. For example, the average pore size is 708.5 μm when the CMC concentration is 0%. However, when the concentration is increased to 0.4%, the average pore size is reduced to 344.2 μm, 48.5% lower than the control group without foam stabilizer. The discharge rate is significantly reduced and, at the same time, inhibits the merging and rupture of bubble liquid film so that the number of small pores increases and the number of large pores decreases. Therefore, it can be concluded that the increase in CMC concentration can sufficiently reduce the average pore size of the pores of foam concrete and has an excellent hole-size adjustment effect.

Figure 10 shows the graph of porosity and average roundness value of foam concrete at different concentrations of CMC. The average roundness value refers to the degree to which the geometry of foamed concrete pores is close to spherical [25]. When the roundness value is close to 1, the hole is spherical. The larger the roundness value is, the more spherical the pore shape is.

Figure 10 shows that the porosity of foam concrete is concentrated between 40% and 65%. The porosity is higher when the CMC concentration is 0%, reaching 64.6%. With the increase in CMC concentration, the porosity presents a downward trend. The porosity decreases to 40.5% at a CMC concentration of 0.4%. However, the porosity increases again to 42.2% at a CMC concentration of 0.5%. The average roundness of foam concrete is mainly between 1.2 and 1.5. The change rule of the average roundness value is mainly divided into the rising stage and the falling stage. The CMC concentration of 0–0.1% is in the rising stage, and the rising rate is relatively gentle. The average roundness value increases from 1.46 to 1.49. The CMC concentration of 0.1–0.5% is the falling stage. The falling rate of the average roundness value in this stage is fast. When the CMC concentration is 0.5%, the average roundness value is 1.25, which is 0.24 lower than 0.1%. Analysis shows that with the increase of the CMC concentration, the porosity and average roundness are reduced, and the pore shape is gradually changed from irregular spherical to regular spherical, effectively improving the pore morphology of foam concrete.

#### 2.2.4. Microscopic Analysis of Foam Concrete

Figure 11 shows the microscopic observation of each group of foam concrete. As shown in Figure 11, the pore size of foam concrete decreases with the increase in CMC concentration. In the C0–C0.3 group, the pore size is large, the distribution among the pores is uneven, and the pore morphology is irregular and circular. In the C0.4–C0.5 group, it is evident that the pore distribution is uniform, the pore wall is dense, the pore size is also significantly reduced, and the pore morphology is nearly circular, which corresponds to the conclusions obtained from the pore structure analysis. In addition, the number of connecting holes in foam concrete also decreases with the increase in CMC concentration. The number of connecting holes in the C0.4–C0.5 group is the same but significantly lower than in the C0 group. The analysis shows that adding CMC can effectively improve the pore structure of foam concrete, restrain the number of large pores and connecting holes, and make the pore size of foam concrete smaller and more uniform.

## 3. Materials and Methods

### 3.1. Test Material

In this experiment, sodium dodecyl sulfate (SDS) (BOSF, Maxam Daily Chemical Co., Ltd. (Shanghai, China)) is used as a foaming agent, sodium carboxymethyl cellulose (CMC) (Feihu Brand, Lihong Fine Chemical Co., Ltd. (Chongqing, China)) is used as foaming stabilizer, and water (Harbin, China) is used to prepare foam. The foam concrete specimens were prepared by adding ordinary Portland cement (P.O. 42.5, Swan brand, Cement Products Co., Ltd. (Harbin, China)), natural river sand (Cement Products Co., Ltd. (Harbin, China)) and polycarboxylic acid superplasticizer (P.C.) (Zhongyan brand, Shunxin Chemical Co., Ltd. (Jinan, China)). The performance indexes of ordinary Portland cement are shown in Table 1.

### 3.2. Sample Mix Ratio and Preparation

This experiment uses the mix ratio of two materials (foaming liquid and foam concrete). The mix ratio of foaming liquid is designed according to the reference [26], as shown in Table 2. The mix ratio of foam concrete is designed by the quality method [27]. The mix ratio of foam concrete is preliminarily determined by controlling the dry density of foam concrete, as shown in Table 3. Then, the sample group is uniformly defined as C0–0.5 according to the different CMC content, where C0 means that the CMC content is 0% and is defined as the control group.

In this test, the foaming agent and concrete foam samples are prepared according to the method specified in JC/T 2199-2013 for foaming agents for foam concrete [28].

The foaming agent preparation process is as follows: according to the set ratio (CMC concentration 0–0.5%, SDS concentration 0.6%), the foaming agent and foam stabilizing agent are slowly mixed into the water and mixed evenly with a glass rod; the foam solution, evenly mixed, is poured into the power-boosting electric stirrer at a speed of 2500 rpm for foaming, and after stirring for 3 min, the foam preparation is completed. 

The preparation process of the foam concrete sample is as follows: (1)Weighing raw materials: According to the above ratio, the corresponding amount of cement, river sand, water-reducing agent, foaming agent, foam stabilizer, and water are weighed accurately with an electronic balance.(2)Preparation of cement slurry: The materials are mixed with a cement mortar mixer (Puls brand, Baoding, Hebei Province, China), and the weighed materials are mixed for 1 min to get the mixed dry mixture. Then the prepared water is added and mixed for 3 min to get the slurry.(3)Prefabricated foam: Power boosting electric agitator (Jj-1, Ningbo, Zhejiang Province, China) is used for physical foaming of foaming agent solution, with a maximum power of 2500 rpm and mixing time of 5 min. The mixing and foaming process of the slurry is carried out at the same time.(4)Foam and slurry mixing: The foam is added to the mixture and stirred until completely uniform. Finally, it is poured into the model of 100 mm × 100 mm × 100 mm brushed with lubricating oil. According to experience, when pouring foamed concrete, it needs to be 2 cm higher than the model due to foam merging and breaking.(5)Scrap, demold, and sample cure: After curing for 24 h, scrape off the excess hardened foam concrete on the surface of the mold with a scraper, demold, and put the test block in the standard curing room (temperature 20 ± 1 °C, relative humidity ≥ 90%) for curing until 28 d.

### 3.3. Experimental Methods

#### 3.3.1. Foam Performance Test

The Waring Blender method [20] is used for the foam volume and half-life test. This method mainly determines low-viscosity aqueous solutions’ foaming ability and foam stability. The operational method is to pour 100 mL of prepared foaming agent as an aqueous solution into a 1 L beaker for foaming and record the foaming volume V by reading the scale of the beaker; next, to record the time required for foam to separate the liquid to 50 mL, test five times, calculate the average value, and record it as the half-life t_1/2_ of the foaming agent as a measure of foam stability. The stirring speed is 2500 RPM, and the mixing time is 5 min.

Following the Chinese standard Technical Code for Design and Construction of Cast in situ foam Light Soil Subgrade (TJG F10 01-2011) [29], the foam bleeding rate test was conducted: the prepared foam was filled into a 50 mL measuring cylinder and observed for 60 min in total, and the volume loss and drop distance of foam was recorded once every 10 min to characterize the stability of foam at specific times under different CMC concentrations. In addition, the digital microscope (GAOSUO, China) was used to conduct microscopic observation on the foam surface 60 min later, at a magnification of 20 times. At the same time, Image software and calculation Formula (1) are used to measure the bubble size and size [30].

GAOSUO digital microscope specifications are as follows:

Magnification: 1×~500×

Imaging distance: manual adjustment 0~infinity

Image resolution: standard 640 × 480

Fixed base: universal metal base (optional lifting bracket)
(1)$$D_{sm}=\frac{\Sigma_{i=1}^{k}D_{i}^{3}}{\Sigma_{i=1}^{k}D_{i}^{2}}$$
where: *Di* is the *i*th bubble size and *k* is the number of bubbles. *D_sm_* was determined by measuring at least 300 bubbles in each photograph.

#### 3.3.2. Foam Concrete Performance Test

The dry density, water absorption, and compressive strength of foam concrete are tested according to the methods specified in the specification [31] of foam concrete (JG/T 266-2011), as shown in Formulas (2)–(4). The size of the foam concrete sample is 100 mm × 100 mm × 100 mm, with three samples for each group. The samples are dried in an electric blast drying oven (Subo, 101, Shaoxing, Zhejiang Province, China), weighed by an electronic balance (Leki, LQ-C5001, Suzhou, Jiangsu Province, China), and tested by an electro-hydraulic press (Hytera, WJB4-TYA-2000, Ningbo, Zhejiang Province, China) for compressive strength, a loading rate of 0.5 KN/s.
(2)$$\rho_{0}=\left(\frac{m_{0}}{\upsilon }\right)$$
where: *ρ*_0_ is the dry density of sample (kg/m^3^); *m*_0_ is the drying mass of the sample (g); *V* is the volume of the sample (mm^3^).
(3)$$W_{R}=\frac{\left(m_{g}-m_{0}\right)}{m_{0}}$$
where: *W_R_* is water absorption rate (%); *m*_0_ is the mass of dried sample (g); *m_g_* is the mass of the sample after water absorption (g).
(4)$$f=\frac{F}{A}$$
where: *f* is the compressive strength of the sample (MPa); *F* is the maximum damage load (KN); *A* is the compressed area of the sample (mm^2^).

The image analysis method [32] analyzes foam concrete pore structure. First, the section photos are binarized through Photoshop software. Then the images are imported into Image Pro Plus (IPP) software to analyze the data related to the pore structure and study the influence of CMC concentration changes on the pore-size distribution, average pore size, porosity, and average roundness of the sample. IPP software operation steps: select Measure—Count/size—Measure—Select Measurements (select the required pore structure parameters: size (mean), Roundness, Per Area, etc.)—Count, then select View—Measurement data to obtain the required hole structure parameters. In addition, a digital microscope (GAOSUO, China) was used to observe the micro section of foam concrete at a magnification of 20 times.

## 4. Conclusions

By adding CMC, a foam stabilizing agent, at different concentrations, the changes in foam properties and foam concrete properties were analyzed. According to the research results, the following conclusions can be drawn:(1)In terms of foam performance, increasing the concentration of CMC in the foaming solution will reduce the foaming volume and affect the foaming ability of the foaming solution. However, the half-life of foam has been significantly improved, and the turning point of slowing down the growth rate of foam half-life is the CMC concentration of 0.4%.(2)The use of CMC can effectively slow the speed of foam merging and bursting, improve the strength of the liquid film, and significantly reduce the amount of foam bleeding. The average pore size of foam changes to become more stable, which is of great significance in improving the stability of the foam.(3)In terms of foam concrete performance, the dry density, water absorption, and compressive strength will change with the increase of CMC concentration, but CMC concentrations that are too high will limit the growth of compressive strength within a specific range. The water absorption of the sample did not increase linearly with the decrease in dry density.(4)With the increase of CMC concentration, the pore size of foamed concrete is reduced, the pore distribution is uniform, and the pore shape is closer to the regular spherical shape, which can effectively improve the pore structure of foamed concrete.

## Figures and Tables

**Figure 1 ijms-23-15473-f001:**
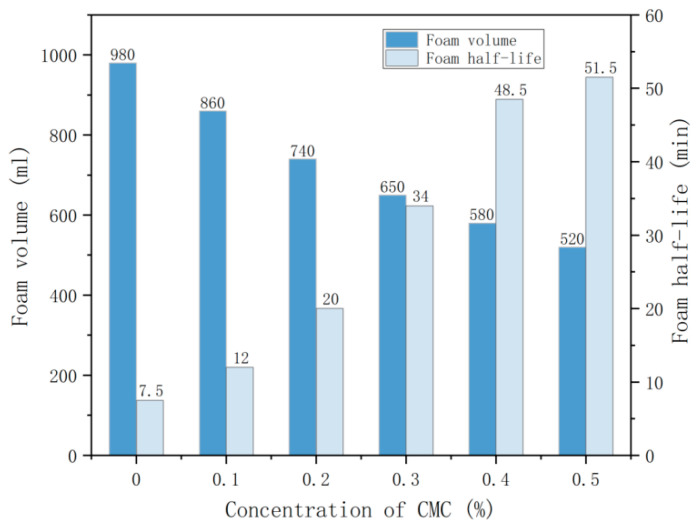
Effect of CMC concentration on foaming volume and half-life of foam.

**Figure 2 ijms-23-15473-f002:**
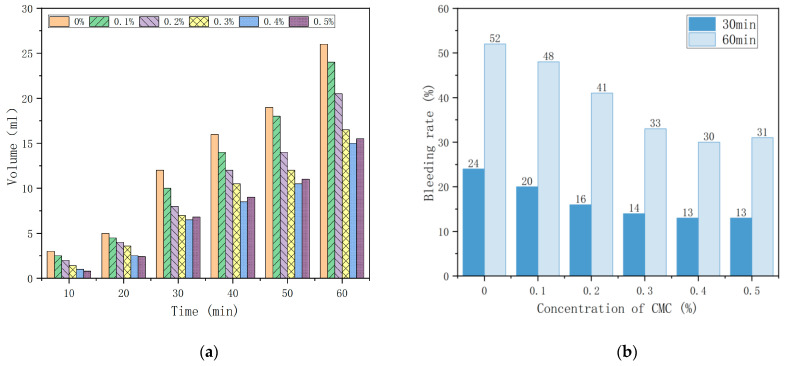
(**a**) Foam bleeding rate in each group as a function of time; (**b**) Foam bleeding rate in 30 min and 60 min in each group.

**Figure 3 ijms-23-15473-f003:**
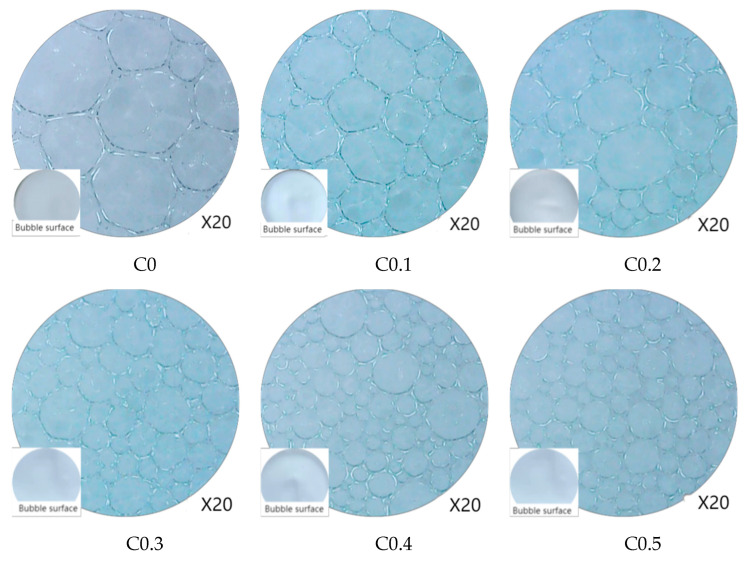
Microscopic pictures of each group at 60 min.

**Figure 4 ijms-23-15473-f004:**
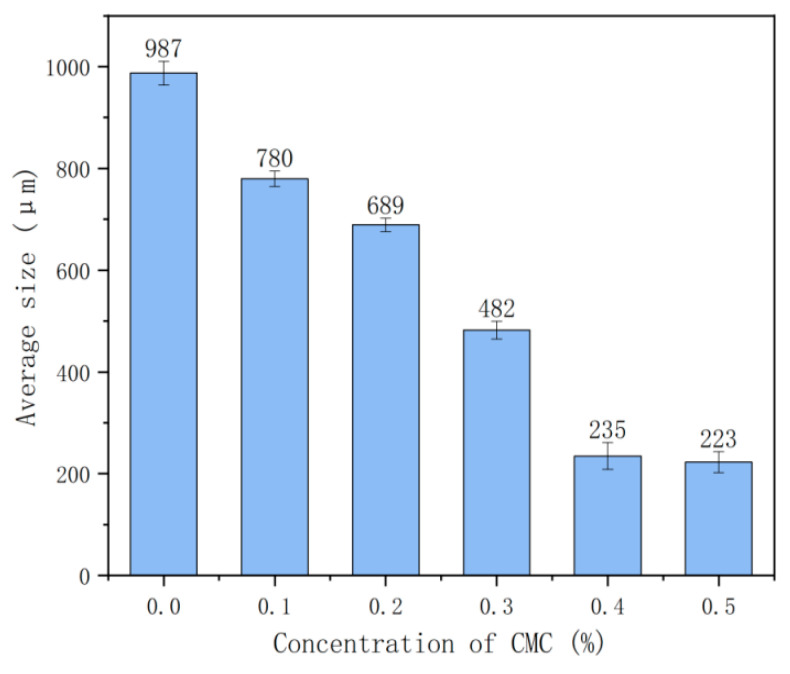
Average size of bubbles in each group at 60 min.

**Figure 5 ijms-23-15473-f005:**
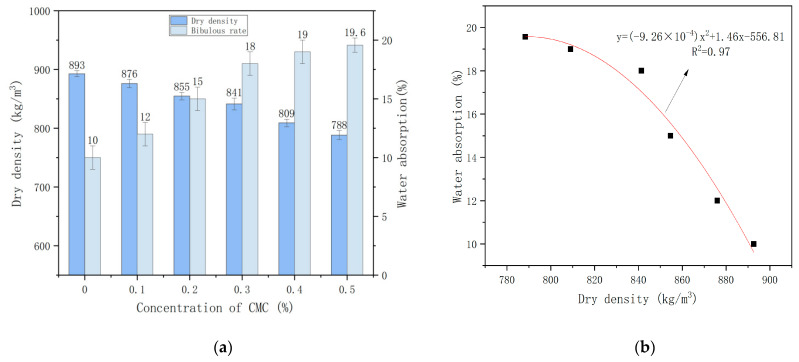
(**a**) Influence of CMC concentration on dry density and water absorption of foam concrete (**b**) Fitting graph of dry density and water absorption.

**Figure 6 ijms-23-15473-f006:**
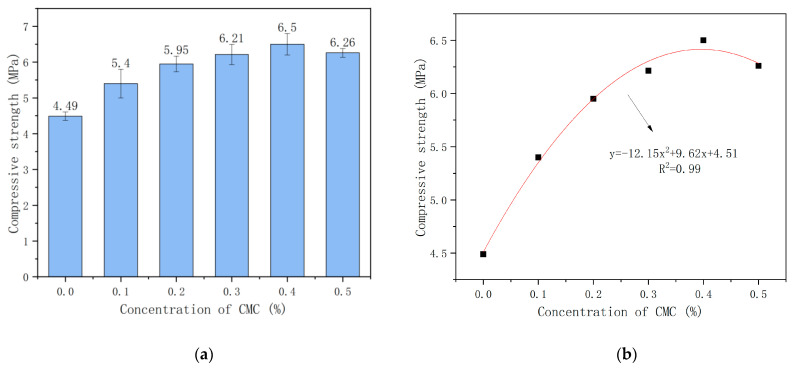
(**a**) Relationship between concentration of CMC and compressive strength; (**b**) Fitting graph of CMC concentration and compressive strength.

**Figure 7 ijms-23-15473-f007:**
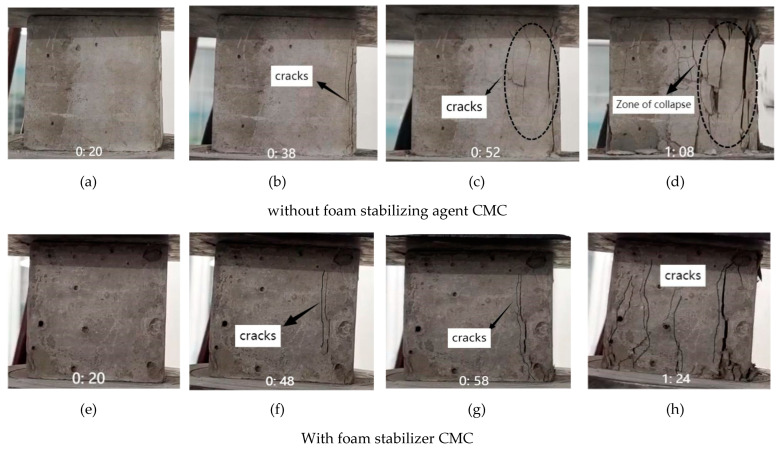
Compression process of foam concrete sample in function of time. Where: 0:20 in the figure refers to the 20th s after the start of compression test.

**Figure 8 ijms-23-15473-f008:**
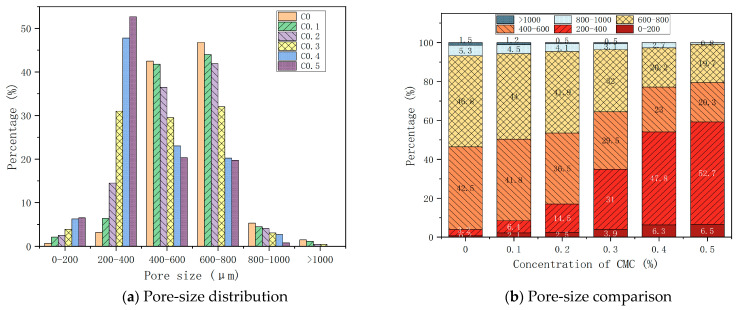
Pore-size distribution at different CMC concentrations used.

**Figure 9 ijms-23-15473-f009:**
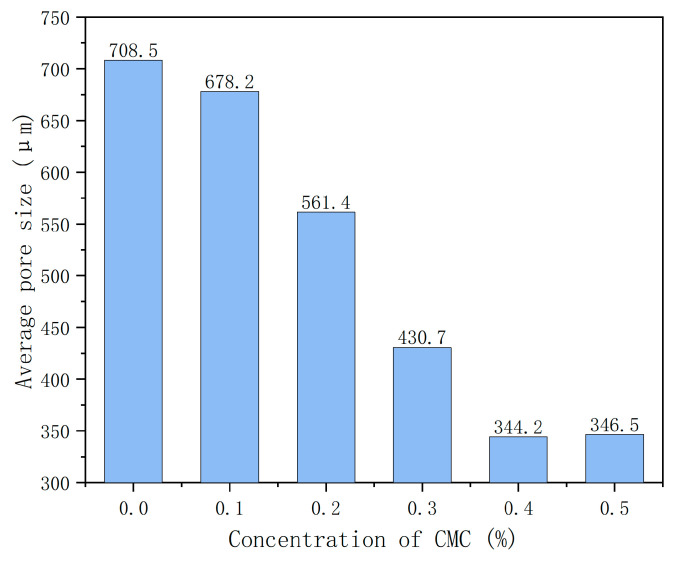
Average pore size at different concentrations of CMC.

**Figure 10 ijms-23-15473-f010:**
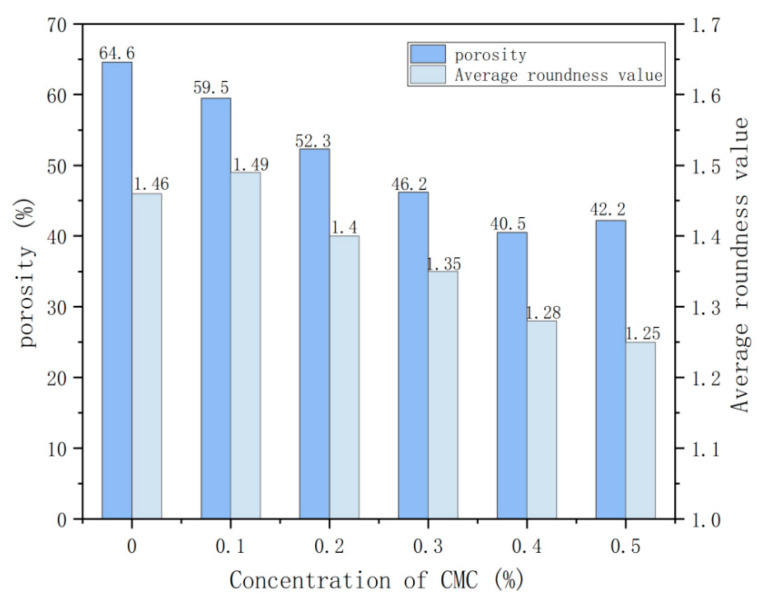
Porosity and average roundness value of foam concrete with different concentrations of CMC.

**Figure 11 ijms-23-15473-f011:**
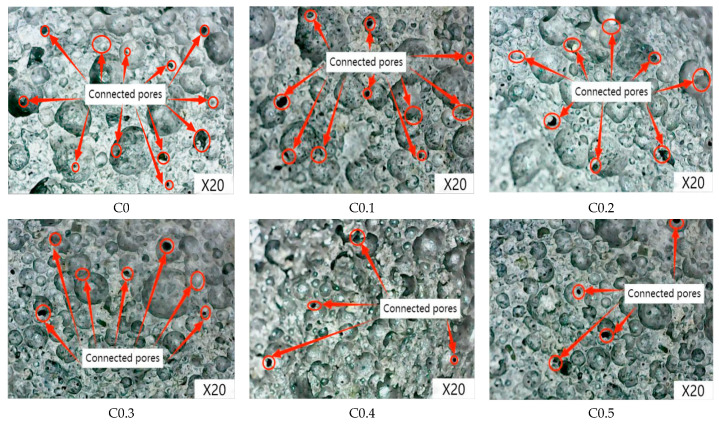
Microscopic observation of each group of foam concrete.

**Table 1 ijms-23-15473-t001:** Performance index of Swan brand P.O. 42.5 silicate cement.

Setting Time/(min)		Compressive Strength/(MPa)		Bending Strength/(MPa)		Water Requirement of Normal Consistency/(%)	Stabilities	Fineness Modulus
Initial Setting	Final Setting	3 d	28 d	3 d	28 d	25.4	qualified	3.2
160	210	25.8	45.2	5.6	9.4			

**Table 2 ijms-23-15473-t002:** Mix ratio of foaming liquid.

Foam Group	SDS/g	CMC/g	Water/g
C0	0.6	0	100
C0.1	0.6	0.1	100
C0.2	0.6	0.2	100
C0.3	0.6	0.3	100
C0.4	0.6	0.4	100
C0.5	0.6	0.5	100

**Table 3 ijms-23-15473-t003:** Mix ratio of foam concrete.

Concrete Group	Target Density Class (kg/m^3^)	Cement (kg/m^3^)	River Sand (kg/m^3^)	Water (kg/m^3^)	Water Reducing Agent (kg/m^3^)	Foam (kg/m^3^)
C0	800	555	111	300	1.3	23.5
C0.1	800	555	111	300	1.3	26.2
C0.2	800	555	111	300	1.3	32.6
C0.3	800	555	111	300	1.3	38.5
C0.4	800	555	111	300	1.3	43.6
C0.5	800	555	111	300	1.3	46.2

## Data Availability

All data generated or analysed during this study are included in this manuscript.

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
