# Peer review of "The Stabilizing Effect of Carboxymethyl Cellulose on Foamed Concrete"

_ijms, 2022, doi:10.3390/ijms232415473_

Round 1

Reviewer 1 Report

Detailed review can be seen in Word document.

Author Response

Dear Reviewer,

The authors thank the reviewers for their manuscript review and valuable suggestions. All comments have been carefully considered and accounted for in a revised manuscript.

thanks,

Yongcheng Ji

Reviewer 2 Report

Veja o arquivo anexado.

Author Response

(The authors gave the same response as above.)

Reviewer 3 Report

Dear Authors,
please, find below several remarks on your interesting manuscript:

-          line 38 – particleboards instead of partition boards?
-          line 42 (table 1) please provide proper units (kg/m3); Drying shrinkage unit?
-          line 86 please correct units (3 in superscript)
-          lines 142, 144, and 146 – add information about material providers
-          line 150 (figure 2) – please add a scale bar
-          line 161 – why you provided kg/m3 unit in the title of the table? Please add units to table columns; are you sure all the data in the table was the same excluding the bubble column?
-          line 181 – where you mentioned Figure 4, in my opinion, Figure 4 is not a chart, but these are two pictures only
-          line 191 – suggests saying stirring speed instead of foaming speed
-          line 212 – please give more information about digital microscope (producer? magnification?)
-          line 216 – I suggest saying “microscopic observation stand”
-          lines 220-221 please list this standard (and all remaining standards) in references
-          line 225 the formula (4) does not require 106 since it depends on what units of data you will use (if you put mass in kg and dimensions in m, then 10^6 is useless)
-          line 235 – these two pictures in Figure 8 are not necessary, no new or special knowledge is provided
-          line 239 – please give the producer of electro-hydraulic press
-          line 263 (Figure 10) – sorry but nothing new in these pictures; more useful to describe the models and producers of the microscopes
-          lines 332-337 I suggest moving to the Methodology section
-          line 340 unnecessary dot between CMC and The
-          line 400 (Figure 16b): please explain why you used an exponential relationship with (R2=0.9) which may suggest that above the concentration of CMC over 0.5 the compressive strength still raise, since, according to the data in Figure 16a there is a decrease of compressive strength for the concentration of CMC = 0.5? Please consider a polynomial relationship, where R2=0.99 (see attached draft plot).

Best regards!

Author Response

(The authors gave the same response as above.)

Round 2

Reviewer 1 Report

Please read in detail published manuscripts on the topic that you studied and write yours in accordance with published ones. 
